# A Novel Soy Isoflavone Derivative, 3′-Hydroxyglycitin, with Potent Antioxidant and Anti-*α*-Glucosidase Activity

**DOI:** 10.3390/plants11172202

**Published:** 2022-08-25

**Authors:** Jiumn-Yih Wu, Tzi-Yuan Wang, Hsiou-Yu Ding, Chuan-Che Lee, Te-Sheng Chang

**Affiliations:** 1Department of Food Science, National Quemoy University, Kinmen County 892, Taiwan; 2Biodiversity Research Center, Academia Sinica, Taipei 11529, Taiwan; 3Department of Cosmetic Science, Chia Nan University of Pharmacy and Science, Tainan 71710, Taiwan; 4Department of Biological Sciences and Technology, National University of Tainan, Tainan 70005, Taiwan

**Keywords:** soy isoflavone, tyrosinase, biotransformation, hydroxylation, antioxidant, glucosidase

## Abstract

This study demonstrated the enzymatic hydroxylation of glycitin to 3′-hydroxyglycitin, confirming the structure by mass and nucleic magnetic resonance spectral analyses. The bioactivity assays further revealed that the new compound possessed over 100-fold higher 1,1-diphenyl-2-picrylhydrazine free-radical scavenging activity than the original glycitin, although its half-time of stability was 22.3 min. Furthermore, the original glycitin lacked anti-*α*-glucosidase activity, whereas the low-toxic 3′-hydroxyglycitin displayed a 10-fold higher anti-*α*-glucosidase activity than acarbose, a standard clinical antidiabetic drug. The inhibition mode of 3′-hydroxyglycitin was noncompetitive, with a K_i_ value of 0.34 mM. These findings highlight the potential use of the new soy isoflavone 3′-hydroxyglycitin in biotechnology industries in the future.

## 1. Introduction

Isoflavones are edible phytoestrogens, natural compounds that accumulate in different plants and are particularly enriched in legumes, such as soybeans [1]. Daidzin, genistin, and glycitin are the main isoflavones in soybeans. In past years, soy isoflavones have been under intensive investigation in hormone-dependent and cardiovascular diseases, breast and prostate cancers, and osteoporosis [2]. Recently, structural modifications of different functional groups in the structure of soy isoflavones have been achieved using genetically engineered microorganisms, resulting in derivatives with dramatically altered bioactivities. 

Among these modifications, hydroxylation of soy isoflavones has attracted much more attention from scientists [3]. For example, Lee et al. used recombinant *Escherichia coli* expressing the tyrosinase of *Bacillus megaterium* (*Bm*TYR) to catalyze the 3′-hydroxylation of soy isoflavone aglycones, daidzein, and genistein in the presence of borate and ascorbic acid [4]. Chang et al. also applied this enzymatic biotransformation system to the catalysis of the 3′-hydroxylation of soy isoflavone glycosides, daidzin, and genistin [5].

In some cases, adding hydroxyl group(s) to soy isoflavone precursors not only increases original bioactivities but also affords novel bioactivities. For example, hydroxylation of soy isoflavones to produce *ortho*-hydroxylated isoflavones greatly increases 1,1-diphenyl-2-picrylhydrazine (DPPH) free-radical scavenging activity compared with that of the original isoflavones [3]. In addition, the *orthro*-hydroxylated isoflavones exhibited other bioactivities, such as anticancer [6,7], antimelanogenesis [8,9], anti-inflammation [10,11], hepatoprotection [12,13], and antitrypanosomal activity [14,15]. A previous study identified certain flavonoids as enzyme inhibitors with potent anti-*α*-glucosidase activity [16]. α-Glucosidase is the major targeted enzyme for antidiabetics because it can cleave oligosaccharides and increase blood sugar concentration. The α-glucosidase inhibitors can block the enzyme to decrease the postprandial blood sugar level and reduce the demand for insulin in diabetes therapy [17,18]. These studies revealed that *Bm*TYR is a promiscuous biocatalyst for the hydroxylation of soy isoflavones. 

In this study, a soy isoflavone, glycitin, was biotransformed by *Bm*TYR into novel functional compounds. The modified glycitin compound was evaluated, purified, and characterized. Both DPPH free radical scavenging activity and anti-*α*-glucosidase activity of the purified derivative were assessed and compared with those of its precursor, glycitin.

## 2. Results and Discussion

### 2.1. Biotransformation of Glycitin by BmTYR

Commercial glycitin was incubated with the *Bm*TYR enzyme in the presence of ascorbate and borate, and the biotransformed products were analyzed using high-performance liquid chromatography (HPLC). The results showed that a major metabolite, compound (**1**), appeared after the biotransformation (Figure 1). As expected, the results revealed that glycitin could be biotransformed by *Bm*TYR [5]. 

To resolve the chemical structure of compound (**1**), the biotransformation was scaled up to 20 mL, and the compound was then purified by preparative HPLC. The chemical structure of purified compound (**1**) was then analyzed using mass and nucleic magnetic resonance (NMR) spectral analyses. The molecular formula of compound (**1**) was established as C_22_H_22_O_11_ by the electrospray ionization mass at m/z 463.3 [M + H]^+^, indicating a molecular weight of 462 (Appendix A). The mass data implied compound (**1**) as 3′-hydroxylglycitin. The distortionless enhancement by polarization transfer (DEPT) spectra exhibited the characteristic compound (**1**) carbon signal at δ: CH_3_ (55.8), CH_2_ (60.6), and CH (69.5, 73.0, 76.7, 77.2, 99.6, 103.3, 104.9, 107.1, 108.8, 118.3, and 152.6). In the aromatic proton spectrum, three signal protons at 8.33 (s), 7.51 (s), and 7.31 (s) were assigned to H-2, H-5, and H-8, respectively, with the remaining three aromatic protons appearing as an A′B′X′ pattern at 6.54 (d, J = 7.7 Hz), 6.76 (dd, J = 7.7, 2.1 Hz), and 6.78 (d, J = 2.1 Hz), assigned to H-2′, H-5′, and H-6′, respectively. The cross-peak of glucosyl H-1″ with C-7 (5.17/151.1 ppm) in the heteronuclear multiple-bond connectivity (HMBC) spectrum demonstrated the structure of compound (**1**). The glycosyl group had a large coupling constant (7.7 Hz) of the anomeric proton H-1″ (δ 5.17 ppm), which indicated the β configuration. Based on the NMR spectral analysis of compound (**1**), the signal for the full assignments of the ^1^H and ^13^C-NMR signals was further aided by DEPT, heteronuclear single quantum coherence (HSQC), HMBC, correlation spectroscopy (COSY), and nuclear Overhauser effect spectroscopy (NOESY) spectra, as shown in Appendix A. These data confirmed that the structure of compound (**1**) was 3′-hydroxylglycitin. Figure 2 illustrates the biotransformation of glycitin by *Bm*TYR. 

In addition, enzymatic hydroxylation of flavonoids could also be achieved by heme-containing cytochrome P450 monooxygenases (P450s), which associate with electron donor partner proteins (cytochrome P450 reductases (CPRs)) to transfer two electrons from the nicotinamide adenine dinucleotide phosphate (NADPH) donor to the P450 heme domain [19]. However, the electron transfer between CPR and P450 proteins usually limits the reaction rate of P450s [19]. Moreover, the high price of the electron donor, NADPH, is another limitation of biotransformation in industrial usage. Without the need for NADPH or CPR, the *Bm*TYR can directly catalyze highly efficient hydroxylation and is easy to scale up [20]. Until now, *Bm*TYR has been proven to catalyze 3′-hydroxylation of daidzein and genistein [4]; daidzin and genistin [5]; liquiritigenin [20]; and glycitin (this study). In these cases, the conversion rates were all greater than 90%. In summary, *Bm*TYR is a valuable biotransformation enzyme for flavonoid hydroxylation and can be applied to other flavonoids in the future.

### 2.2. 3′-Hydroxylglycitin Possesses Potent Antioxidant Activity 

According to a previous study, the *ortho*-dihydroxyl groups on the benzene ring of flavonoid structures play an important role in their antioxidant activities [21]. Thus, the antioxidative activities of both 3′-hydroxylglycitin and its precursor glycitin were determined by a DPPH free radical scavenging assay. The results showed that the antioxidant activity of 3′-hydroxylglycitin (IC_50_ = 134.7 ± 8.7 μM) was over 100-fold higher than that of glycitin (IC_50_ >> 8 mM) and comparable to that of ascorbic acid (IC_50_ = 111.0 ± 5.6 μM) (Figure 3). This remarkable antioxidant activity of 3′-hydroxylglycitin may confer bioactivities, such as skin whitening, anticancer, and antidiabetic activities [3].

Some potent antioxidants, such as common ascorbic acid and vitamin E, are unstable due to spontaneous oxidation. Thus, the stability of 3′-hydroxyglycitin was evaluated. The results confirmed that the potent antioxidant, 3′-hydroxyglycitin, was unstable with a half-time of 22.3 min at the tested condition (Figure 4). 

### 2.3. 3′-Hydroxylglycitin Possesses Potent Anti-α-Glucosidase Activity 

Some natural flavonoids were identified as having potent anti-*α*-glucosidase activity [16]. Therefore, the anti-*α*-glucosidase activity of both glycitin and 3′-hydroxylglycitin was also investigated. Acarbose is a standard α-glucosidase inhibitor, which is a common antidiabetic drug used as a positive standard in the assay. The results showed that 3′-hydroxylglycitin (IC_50_ = 0.36 ± 0.02 mM) possessed 10-fold higher anti-*α*-glucosidase activity than acarbose (IC_50_ = 3.47 ± 0.23 mM) (Figure 5). By contrast, no anti-α-glucosidase activity of glycitin was observed under the tested conditions. A kinetic study of α-glucosidase inhibition was performed. The results showed that 3′-hydroxylation was a noncompetitive inhibitor of *α*-glucosidase with a Ki value of 0.34 ± 0.01 mM (Figure 6). These results demonstrated that the enzymatic hydroxylation of glycitin into 3′-hydroxylglycitin resulted in a derivative with a highly improved antioxidant property (Figure 3) and novel anti-*α*-glucosidase activity (Figure 5). 

The cytotoxicity of the compound was evaluated using mouse B16 melanoma cells because the potent anti-*a*-glucosidase activity of 3′-hydroxyglycitin shows promise an antidiabetic drug. The results showed that 3′-hydroxyglycitin was nontoxic to the cells even at a 0.4 mM concentration (Figure 7). The low-toxicity and potent anti-*a*-glucosidase activity of 3′-hydroxyglycitin may be applied to medicine and/or healthcare in the future.

## 3. Materials and Methods

### 3.1. Cells, Chemicals, and Microorganisms

Mouse B16-F10 melanoma cells BCRC 60031 were bought from the Bioresources Collection and Research Center (BCRC, Food Industry Research and Development Institute, Hsinchu, Taiwan). *Bm*TYR with a specific activity of 4.84 U/mg was obtained from a previous study [20]. Glycitin was purchased from Baoji Herbest Bio-Tech (Xi’An, Shaanxi, China). DPPH, *α*-glucosidase from *Saccharomyces cerevisiae* (G5003-100UN, 10 U/mg), *p*-nitrophenyl-*α*-d-glucoside (PNPG), 3-(4,5-dimethylthiazol-2-yl)-2,5-diphenyltetrazolium bromide (MTT), dimethyl sulfoxide (DMSO), l-dihydroxyphenylalanine (l-DOPA), and ascorbic acid were purchased from Sigma (St. Louis, MO, USA). The other reagents and solvents used were commercially available. 

### 3.2. Biotransformation Using BmTYR

The biotransformation system was operated according to the method reported by Lee et al. [4], with minor modifications. The reaction mixture (100 μL) containing 500 mM of borate (pH 9.0), 10 mM of ascorbic acid, 1 mg/mL of the tested substrate compound (diluted from a stock of 20 mg/mL in DMSO), and 108 μg/mL of *Bm*TYR was incubated at 50 °C and shaken at 200 rpm for 1.5 h. At the end of the reaction, 20 μL of 1 M HCl and 120 μL of MeOH were added to stop the reaction, and the sample was analyzed using HPLC.

### 3.3. HPLC Analysis

HPLC was performed with an Agilent 1100 series HPLC system (Santa Clara, CA, USA) equipped with a gradient pump (Waters 600, Waters, Milford, MA, USA). The stationary phase was a C18 column (5 μm, 4.6 i.d. × 250 mm; Sharpsil H-C18, Sharpsil, Beijing, China), and the mobile phase was 1% acetic acid in water (A) and methanol (B). The elution condition was a linear gradient from 0 min with 20% B to 20 min with 50% B, an isocratic elution from 20 min to 25 min with 50% B, a linear gradient from 25 min with 50% B to 28 min with 20% B, and an isocratic elution from 28 min to 35 min with 20% B. All eluants were eluted at a flow rate of 1 mL/min. The sample volume was 10 μL. The detection condition was set at 254 nm.

### 3.4. Purification and Identification of the Biotransformation Metabolite 

The purification process was a previously described method [20]. To purify the biotransformation metabolite compound (**1**), the biotransformation reaction was scaled up to 20 mL. After the large-scale reaction, an equal volume of methanol was added to stop the biotransformation. The mixture was then filtrated with a 0.2 μm nylon membrane, and the filtrate was injected into a preparative YoungLin HPLC system (YL9100, YL Instrument, Gyeonggi-do, South Korea) equipped with a preparative C18 reversed-phase column (10 μm, 20.0 i.d. × 250 mm, ODS 3; Inertsil, GL Sciences, Eindhoven, The Netherlands). The operational conditions for the preparative HPLC analysis were the same as those in the analytic HPLC analysis. The elution corresponding to the peak of the metabolite in the analytic HPLC analysis was collected, condensed under a vacuum, and then crystallized by freeze-drying. Eventually, 17.4 mg of compound (**1**) was obtained. Based on the results HPLC analysis, the purity of the purified compound (**1**) was estimated as 96.7%, and the yield of the purified compound (**1**) was 85.6%. The structure of the compound was confirmed by NMR and mass spectral analyses. Mass analysis was performed using a mass spectrometer (AB Sciex Instruments QTRAP 5500, Applied Biosystem Corp., Foster, CA, USA) with electrospray ionization (ESI). Mass spectrometry was performed using the following parameters: capillary voltage, 4.5–5.5 kV; desolvation gas pressure, 20 psi. Mass signals were collected using a single-ion recording method and processed using Analyst 1.5 software (Applied Biosystem Corp.). Full mass-scan and mass-mass scan data were acquired with a mass range of 100 to 1150 m/z in positive ion mode. For NMR analysis, 5 mg of 3′-hydroxyglycitin was dissolved in 0.5 mL of DMSO-d6 in a 5 mm diameter NMR tube. We recorded 1D NMR (^1^H, ^13^C, and DEPT) and 2D NMR (HSQC, HMBC, COSY, and NOESY) spectra on a high-resolution NMR spectrometer (700 MHz, AVANCE 700, Bruker Company, Billerica, MA, USA) at ambient temperature. Standard pulse sequences and parameters were used for the NMR experiments, and all chemical shifts are reported in parts per million (ppm, *δ*).

Compound (**1**): light yellow powder; ESI/MS m/z: 463.3 [M + H]^+^, 301.1. ^1^H-NMR (DMSO-*d_6_*, 700 MHz) H*δ*: 3.18 (1H, m, H-4″), 3.31 (1H, m, H-2″), 3.33 (1H, m, H-3″), 3.46 (1H, m, H6″a), 3.48 (1H, m, H-5″), 3.70 (1H, m, H-6″b), 3.89 (3H, s, OCH_3_), 5.17 (1H, d, *J* = 7.7 Hz, H-1″), 6.54 (1H, d, *J* = 7.7 Hz, H-5′), 6.76 (1H, dd, *J* = 7.7, 2.1 Hz, H-6′), 6.78 (1H, d, *J* = 2.1 Hz, H-2′), 7.31 (1H, s, H-8), 7.51 (1H, s, H-5), 8.33 (1H, s, H-2). ^13^C-NMR (DMSO-*d_6_*, 175 MHz) C*δ*: 55.8 (OCH_3_), 60.6 (C-6″), 69.5 (C-4″), 73.0 (C-2″), 76.7 (C-3″), 77.2 (C-5″), 99.6 (C-1″), 103.3 (C-8), 104.9 (C-5), 107.1 (C-5′), 108.8 (C-2′), 117.9 (C-4a), 118.3 (C-6′), 121.1 (C-1′), 124.4 (C-3), 147.3 (C-6), 151.1 (C-7), 151.2 (C-3′), 151.3 (C-4′), 151.6 (8a), 152.6 (C-2), 174.6 (C-4)

### 3.5. Determination of Antiradical Activity Using a DPPH Assay 

The assay was performed as previously described, with minor modifications [20]. The tested sample (dissolved in DMSO) was added to the DPPH solution (1 mM in methanol) to a final volume of 0.1 mL. After 15 min of reaction, the absorbance of the reaction mixture was measured at 517 nm using a microplate reader (Sunrise, Tecan, Männedorf, Switzerland). Ascorbic acid (dissolved in DMSO) was used as a positive antioxidant standard. 

### 3.6. Determination of Stability

The stability assay was evaluated as previously reported [22]. A fresh stock of glycitin and 3′-hydroxyglycitin (20 mg/mL in DMSO) was diluted 20-fold to 1 mL with a final concentration of 1 mg/mL in 50 mM of PB buffer at pH 6.8. The diluted solutions in 1.5 mL tubes covered with alumni fossils were placed at 25 °C for 120 min in the dark. During the testing time, 50 μL was aliquoted for the HPLC analysis at the determined times.

### 3.7. Determination of Anti-α-Glucosidase Activity

The conditions for the assay of the yeast *α*-glucosidase inhibitory activity were performed in a 96-well plate. The procedure was as follows: 49 μL of 50 mM PB buffer (pH 6.8) containing 0.2 U/mL of the yeast *α*-glucosidase (dissolved in 50 mM PB buffer (pH 6.8) with 20% (*w/v*) of sorbitol) was mixed with 2 μL of the tested compound (dissolved in DMSO) at 25 °C for 2 min. Then, 49 μL of 5 mM enzyme substrate PNPG (dissolved in PB buffer) was added to the mixture. The absorbance at 405 nm of the liberated *p*-nitrophenol was monitored every 5 s for 2 min with a microplate reader (Sunrise, Tecan, Männedorf, Switzerland). The concentration of an inhibitor required to inhibit 50% enzyme activity under the assay conditions was defined as the IC_50_ value. In the kinetic study, different concentrations of PNPG (0.625, 1.25, 2.5, and 5 mM) were used. Kinetic parameters were obtained from the double-reciprocal plot of substrate PNPG concentration versus the rate of reaction. The kinetic parameters were calculated by nonlinear regression analysis applied to the Michaelis–Menten equation using SigmaPlot 14.5 software (Systat Software, San Jose, CA, USA).

### 3.8. Cytotoxicity by Determination of Cell Viability

Cell viability was determined as previously reported [22] and is briefly described below. Dulbecco’s modified Eagle’s medium (DMEM) containing 10% (*v/v*) fetal bovine serum was used to cultivate the mouse B16 melanoma cells, which were incubated at 37 °C in a humidified, 5% CO_2_ incubator for one day. The cells were then treated with tested drugs for another 2-day incubation. The treated cells were then harvested for the MTT assay as previously described [22].

## 4. Conclusions

Glycitin was biotransformed by *Bm*TYR to produce a new compound, 3′-hydroxyglycitin. The DPPH free radical scavenging activity of 3′-hydroxyglycitin was over 100-fold higher than that of glycitin and comparable to that ascorbic acid, although 3′-hydroxyglycitin was unstable. In addition, 3′-hydroxyglycini possessed 10-fold higher anti-*α*-glucosidase activity than an antidiabetic drug (acarbose), and its inhibition mode was noncompetitive, with a K_i_ value of 0.34 mM. Moreover, 3′-hydroxyglycitin showed noncytotoxicity toward mouse B16 cells even at 0.4 mM. The new soy isoflavone glycoside may be applied to biotechnology in the future.

## Figures and Tables

**Figure 1 plants-11-02202-f001:**
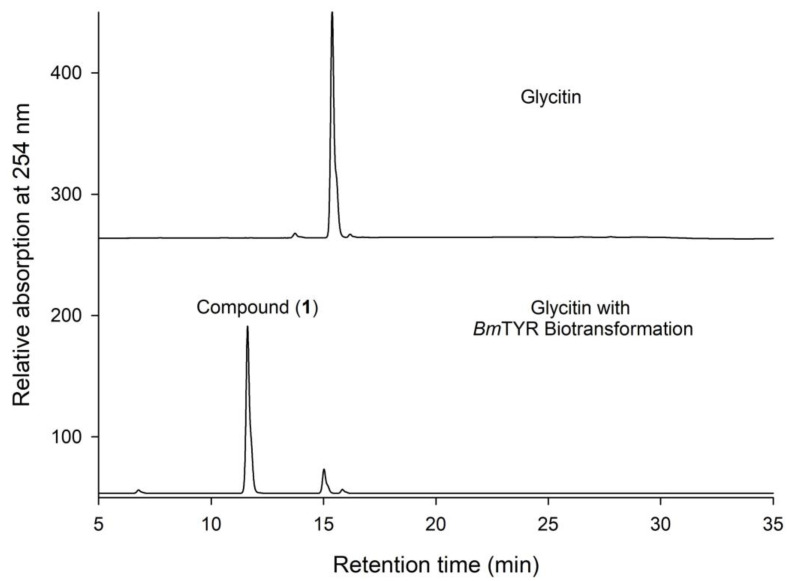
High-performance liquid chromatography (HPLC) analysis of the biotransformation products of glycitin using *Bm*TYR. The biotransformation mixture containing 108 μg/mL of the purified recombinant *Bm*TYR enzymes, 1 mg/mL of glycitin, 10 mM of ascorbic acid, and 500 mM borate buffer at pH 9 was incubated at 50 °C and 200 rpm shaking for 1.5 h. At the end of the reaction, a one-fifth volume of 1 M HCl and an equal volume of MeOH were added to stop the reaction, and the sample was analyzed using HPLC. The HPLC operation procedure is described in the Section 3.

**Figure 2 plants-11-02202-f002:**
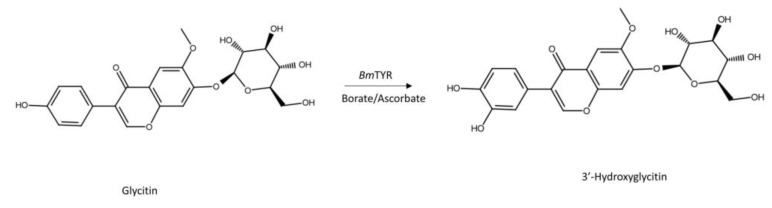
The biotransformation process of glycitin by *Bm*TYR.

**Figure 3 plants-11-02202-f003:**
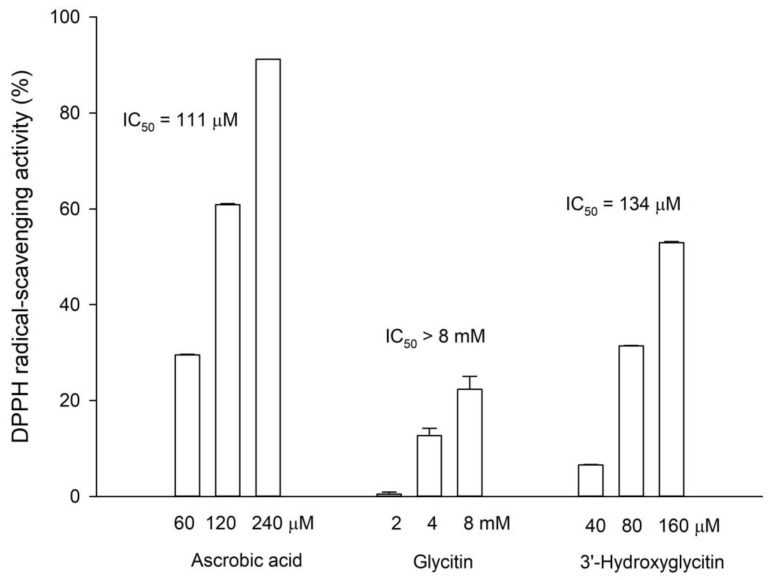
DPPH free radical scavenging activity of ascorbic acid, glycitin, and 3′-hydroxyglycitin. The DPPH free radical scavenging activity was calculated as follows: DPPH free radical scavenging activity = (OD_517_ of the control reaction - OD_517_ of the reaction)/(OD_517_ of the control reaction). The concentration of an inhibitor required to scavenge 50% of the initial DPPH free radical under the assay conditions was defined as the IC_50_ value. The mean (*n* = 3) is shown, and the standard deviation is represented by the error bar. The IC_50_ values represent the concentrations required for 50% DPPH free radical scavenging activity.

**Figure 4 plants-11-02202-f004:**
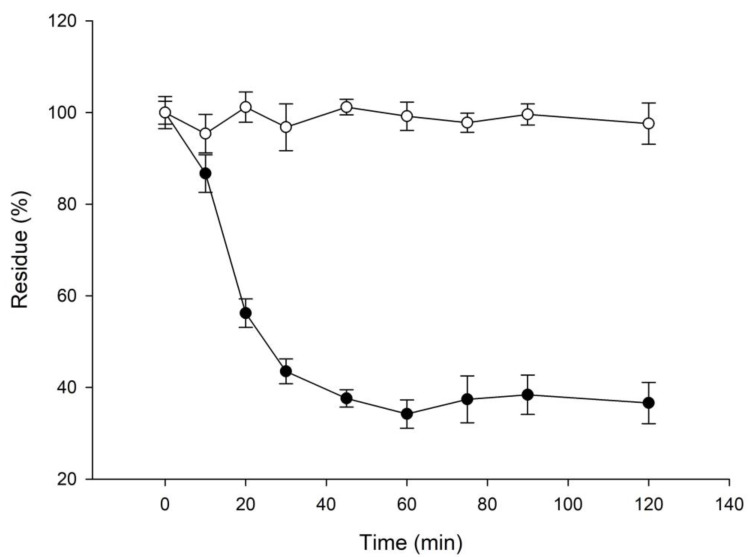
Stability of glycitin (open circle) and 3′-hydroxyglycitin (closed circle). The stability was analyzed by a HPLC method described in the Materials and Methods. The mean (*n* = 3) is shown, and the standard deviations are represented by error bars.

**Figure 5 plants-11-02202-f005:**
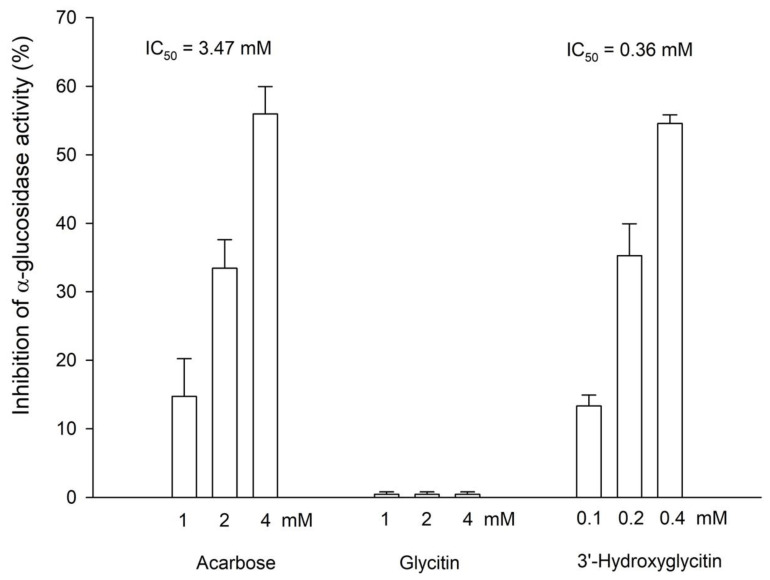
Inhibition of *a*-glucosidase activity by acarbose, glycitin, and 3′-hydroxyglycitin. The initial rate (R_i_) was recorded from the slope of the linear plot of absorbance versus time in each reaction. Acarbose was used as a positive control, and DMSO was used as a negative control. The inhibition of enzyme activity was calculated as follows: Inhibitory effect (%) = (R_i_ of negative control − R_i_ of the tested compound)/R_i_ of negative control × 100. The mean (*n* = 3) is shown, and the standard deviation is represented by the error bar. The IC_50_ values represent the concentrations required for 50% inhibition activity.

**Figure 6 plants-11-02202-f006:**
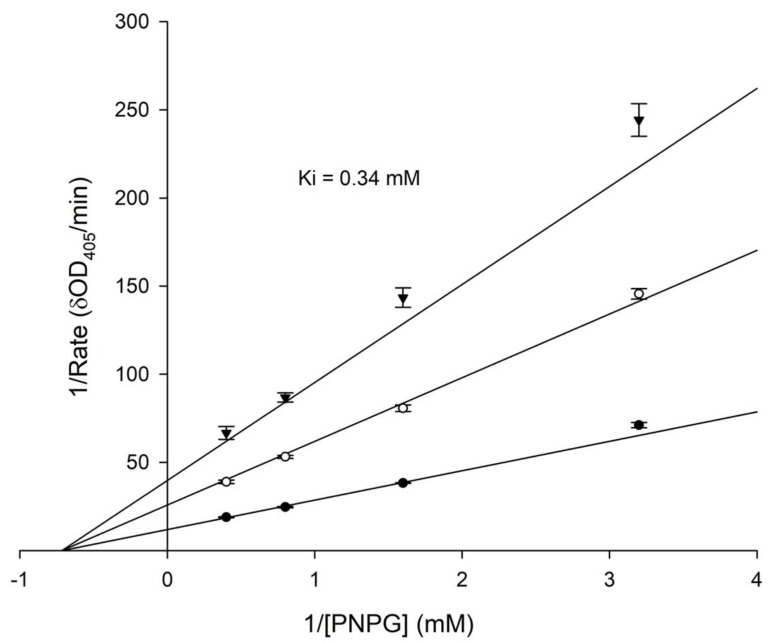
Lineweaver–Burk plots of *a*-glucosidase and *p*-nitrophenyl-*α*-d-glucoside (PNPG) without (●) and with 0.4 mM (▼) and 0.2 mM (○) of 3′-hydroxyglycitin. The *α*-glucosidase activity was determined as described in the Materials and Methods. The mean (*n* = 3) is shown, and the standard deviation is represented by the error bar.

**Figure 7 plants-11-02202-f007:**
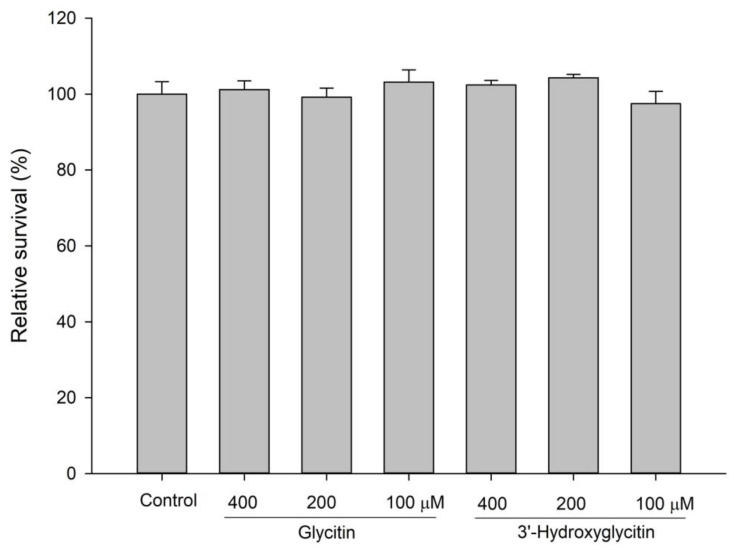
Cytotoxicity of glycitin and 3′-hydroxyglycitin on mouse B16 melanoma cells. The cells were seeded in 24-well plates for one day and then treated with various dosages of the tested drugs for another two days. Cell viability was then examined by an MTT assay. The mean (*n* = 3) is shown, and the SD is represented by error bars.

## Data Availability

Not applicable.

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
