# Peer review of "A Novel Soy Isoflavone Derivative, 3′-Hydroxyglycitin, with Potent Antioxidant and Anti-α-Glucosidase Activity"

_plants, 2022, doi:10.3390/plants11172202_

Round 1
Reviewer 1 Report
Thank you for the opportunity to review the article “A novel soy isoflavone derivative, 3’-hydroxyglycitin, with potent anti-oxidant and anti-α-glucosidase activity”. The manuscript mainly investigated a soy isoflavone, glycitin, was biotransformed by BmTYR into novel functional compounds. The modified glycitin compound was evaluated, purified, and characterized. Both DPPH free radical scavenging activity and anti-α-glucosidase activity of the purified derivative were assessed and compared with those of its precursor, glycitin. The objective is clear. The paper is well written, easy to follow and persuasive, methodology is sound. However, there are some comments and suggestions for the authors in order to improve the quality of their manuscript:
1. Lines 13-17. “Soy isoflavones, including daidzin, genistin, and glycisin, are major constituents of soybeans and possess many bioactivities. Our previous study showed that both daidzin and genistin could be hydroxylated by Bacillus megaterium tyrosinase (BmTYR) to produce relative 3’-hydroxyl derivatives. Based on the similarities in the structures between soy isoflavones, we speculated that glycitin might be hydroxylated to an expected new compound by BmTYR.”. This manuscript focused on the antioxidant and anti-α-glucosidase activity of the modified glycitin. It is recommended to abbreviate or delete this sentence in the abstract.
2. Why choose glycitin for hydroxylation? What about other isoflavones?
3. The DPPH free radical scavenging ability has been analyzed here, what are the other antioxidant abilities of 3'-hydroxy-glycitin, such as ABTS?
4. Glycitin could be biotransformed by BmTYR to produce a new compound, 3’-hydroxy-glycitin, which possessed potent DPPH free radical-scavenging activity and remarkable α-glucosidase inhibitory activity. How is the stability of 3’-hydroxy-glycitin? Is it easy to oxidatively decompose, or does it have other toxicity? Please supplemental the stability and toxicity experiments.
5. The biotransformation products of glycitin using BmTYR is analyzed by HPLC. What is the purity and extraction yield of 3'-hydroxy-glycitin after purification?
6. The error bar in the DPPH free radical scavenging ability of 2 mM glycitin (Figure 3) is too large, and it is recommended to re-measure.
7. Please check and modify the format of the manuscript according to the journal requirements.
Author Response
Please see the attached pdf file.

Reviewer 2 Report
Minor remarks
Provide a blank space between quantity and Celsius degree.
Please, avoid using the third-person singular in the scientific paper.
Use the italic letters for Greek symbols. Please, check it in the whole manuscript.
The abbreviation should be defined after the first mention in the paper.
Major remarks
A better literature review and comparison of the obtained results with literature data should be given in the manuscript. The hydroxylation of isoflavones using biotransformation should be better discussed and explained the advantages of this approach compared with other used methods. Are there other methods?
More detail about MS and NMR analysis should be provided. Please, be specific when you mention the ESI conditions.
The conclusion should be extended and represents the main conclusions obtained after detailed research. Also, the main contribution of the study should be presented.

Author Response
Please see the attached pdf file.

Round 2
Reviewer 1 Report
The authors carefully studied the review comments and gave the scientific replies. At the same time, the author revised the manuscript according to the review comments. I think the revised paper can be accepted and published by the journal.